# Impact of COVID-19 vaccination on liver transplant recipients. Experience in a reference center in Mexico

Daniel Azamar-Llamas[1], Josealberto Sebastiano Arenas-Martinez[1], Antonio Olivas-Martinez[2]*, Jose Victor Jimenez[3,4], Eric Kauffman-Ortega[1], Cristian J García-Carrera[1], Bruno Papacristofilou-Riebeling[1], Fabián E Rivera-López[1], Ignacio García-Juárez[1]

1 Department of Gastroenterology and Liver Transplant Unit, Instituto Nacional de Ciencias Médicas y Nutrición Salvador Zubirán, Mexico City, Mexico, 2 Department of Biostatistics, University of Washington, Seattle, Washington, United States of America, 3 Department of Internal Medicine, Instituto Nacional de Ciencias Médicas y Nutrición Salvador Zubirán, Mexico City, Mexico, 4 Department of Internal Medicine, Yale New Haven Hospital, New Haven, Connecticut, United States of America

* antonio.olivas@gmail.com

## Abstract

### Background and aims

COVID-19 vaccination has proved to be effective to prevent symptomatic infection and severe disease even in immunocompromised patients including liver transplant patients. We aim to assess the impact of COVID-19 vaccination on the mortality and development of severe and critical disease in our center.

### Methods

A retrospective cohort study of LT patients in a reference center between March 2020 and February 2022. Demographic data, cirrhosis etiology, time on liver transplantation, immuno-suppressive therapies, and vaccination status were recorded at the time of diagnosis. Primary outcome was death due to COVID-19, and secondary outcomes included the development of severe COVID-19 and intensive care unit (ICU) requirement.

### Results

153 of 324 LT recipients developed COVID-19, in whom the main causes of cirrhosis were HCV infection and metabolic-associated fatty liver disease. The vaccines used were BNT162b2 (48.6%), ChAdOx1 nCoV-19 (21.6%), mRNA-1273 vaccine (1.4%), Sputnik V (14.9%), Ad5-nCoV-S (4.1%) and CoronaVac (9.5%). Case fatality and ICU requirement risk were similar among vaccinated and unvaccinated LT patients (adjusted relative case fatality for vaccinated versus unvaccinated of 0.68, 95% CI 0.14–3.24, p = 0.62; adjusted relative risk [aRR] for ICU requirement of 0.45, 95% CI 0.11–1.88, p = 0.27). Nonetheless, vaccination was associated with a lower risk of severe disease (aRR for severe disease of 0.32, 95% CI 0.14–0.71, p = 0.005).

**Data Availability Statement:** All relevant data are within the manuscript.

**Funding:** The author(s) received no specific funding for this work.

**Competing interests:** The authors have declared that no competing interests exist.

**Abbreviations:** ARDS, Acute Respiratory Distress Syndrome; aRR, adjusted relative risk; BMI, body mass index; COVID-19, Coronavirus infectious disease 19; DM, diabetes mellitus; HCV, hepatitis C virus; ICU, intensive care unit; LT, liver transplant; NIH, National Institutes of Health; RT-PCR, reverse transcriptase polymerase chain reaction; SARS-CoV-2, severe acute respiratory syndrome coronavirus 2.

## Conclusions

Vaccination reduces the risk of severe COVID-19 in LT patients, regardless of the scheme used. Vaccination should be encouraged for all.

## 1. Introduction

Mexico has one of the highest COVID-19 related excess mortality globally, with a rate of 325.1 deaths per 100,000 population, fourth only to India, the USA, and Russia [1]. The high mortality is attributable to various factors, including the saturation of intensive care unit (ICU) beds, low availability of ICU resources, conversion of general wards into enabled ICU facilities [2] inadequate expertise among healthcare personnel in treating severe ARDS during the initial stages of the pandemic [3], and the high prevalence of metabolic conditions such as diabetes mellitus and obesity [4].

Liver transplant (LT) patients do not appear to be more prone to COVID-19 infection with a similar incidence rate than the general population (3.18 cases/100 person-years in LT patients vs. 3.97 cases/100 person-years in non-LT patients), also featuring lower in-hospital mortality (18% vs. 27%) [5–8]. Vaccination has shown a protective effect against COVID-19, and to severe disease and death, irrespective of the vaccination scheme in the general population [9]. Patients with liver transplantation have a blunted humoral immune response to hepatitis A, hepatitis B, pneumococcal, and COVID-19 vaccination due to immunosuppressive therapy, however, in the latter the clinical significance of this is still an area of research [10–16]. In a case series of 19 LT individuals from an international registry, patients with a complete vaccination scheme against SARS-CoV-2 (12/19) had fewer severe cases and no deaths compared to partially vaccinated and unvaccinated patients. These findings suggest that COVID-19 vaccination may protect LT patients against severe infection, especially when receiving a full vaccine regimen [16].

This study aimed to describe COVID-19 cases among LT patients at a tertiary care center in Mexico City and to assess the impact of COVID-19 vaccination on major outcomes, such as death due to COVID-19, progression to severe disease, and ICU requirement, among LT patients who develop symptomatic COVID-19.

## 2. Methods and materials

### 2.1 Ethics statement

The study was conducted according to the principles of the Declaration of Helsinki (World Medical Association Declaration of Helsinki Ethical Principles for Medical Research Involving Human Subjects, Version Fortaleza, Brazil, October 2013,). This research was conducted in full compliance with the ethical guidelines and regulations set forth by the institutional research and ethics committee (*Comité de Investigación del Instituto Nacional de Ciencias Médicas y Nutrición Salvaodr Zubiran* and *Comité de Ética en Investigación del Insituto Nacional de Ciencias Médicas y Nutrición Salvador Zubiran*, registration number 09-CEI-011-20160627, approval number: REF. 3678, date of approval April 12th 2021). As information was analyzed anonymously and the data was extracted from electronic medical records the consent was waived by the institutional research and ethics committee.

## 2.2 Study design and patients

This is a retrospective cohort study including all LT patients with COVID-19 who were assessed in a tertiary care center in Mexico City between March 1st 2020 and February 28th 2022. The diagnosis of COVID-19 was established if the individual had symptoms compatible with COVID-19 and a positive reverse transcriptase polymerase chain reaction (RT-PCR) test of a nasal or pharyngeal swab sample. The data was extracted from the electronic medical records from October 1st 2021 to March 31st 2022, the authors couldn't identify individual participants during or after data collection.

## 2.3 Variables and definitions

Demographic characteristics (age, sex, body mass index [BMI], etiology of liver disease, hepatocarcinoma, diabetes, hypertension), cirrhosis etiology, time on liver transplantation, immunosuppressive therapies, and vaccination status were recorded at the time of COVID-19 diagnosis. A patient was classified as vaccinated if he or she had a complete scheme with or without a booster. The vaccines used were BNT162b2, ChAdOx1 nCoV-19, mRNA-1273, Sputnik V, and CoronaVac as two-dose schemes and Ad5-nCoV-S as a one-dose scheme, according to national policies. Patients who became infected within 14 days after completing the scheme were considered unvaccinated. The information was collected from electronic medical records.

## 2.4 Outcomes

The primary outcome was death due to COVID-19. Secondary outcomes were the development of severe COVID-19 disease and ICU requirement. Death due to COVID-19 was defined as a death during hospitalization and with COVID-19 as its immediate or underlying cause of death in the death certificate. Patients with COVID-19 were considered to have severe illness if they had SpO2 <94% on room air at sea level, PaO2/FiO2 <300 mm Hg, respiratory rate >30 breaths/min, or lung infiltrates >50%. ICU requirement was defined as patients who developed severe acute respiratory distress syndrome (ARDS) by Berlin Criteria, acute respiratory failure with refractory hypoxemia (PaO2 <55mmHg with supplemental oxygen), septic shock, or multiple organ dysfunction.

## 2.5 Statistical analysis

Demographic and liver transplant-related characteristics are described overall and stratified by vaccination status. Numerical variables are summarized with either mean and standard deviation (SD) or median and interquartile range (IQR) as appropriate and compared between vaccinated and unvaccinated groups using the t test that allows unequal variances. Categorical variables are presented in counts and percentages and compared with the chi-square test or Fisher's exact test as appropriate.

The relationship between vaccination and each outcome was assessed with a Poisson regression model that included vaccination status as the exposure of interest and that used robust standard error estimates (unadjusted analysis). Additionally, similar regression models were fitted but also adjusting for age and time on liver transplantation shorter than 6 months (adjusted analysis). Age was considered as a confounder since it is related to vaccination status due to the national vaccination policy and to the outcomes. A time from liver transplantation shorter than 6 months was considered as a precision variable since LT patients are more expected to be immunocompromised within 6 months after LT and are therefore at a higher risk of developing severe infections.

Finally, the number of vaccinated and unvaccinated LT patients, as well as the number of COVID-19 cases, severe cases, and deaths due to COVID-19 are displayed over time to visualize the relationship between vaccination and COVID-19 outcomes.

## 3. Results

### 3.1 Demographic and clinical characteristics at the time of infection

Between March 2020 and February 2022, 153 cases of COVID-19 were identified among LT patients. Their demographic and clinical characteristics at the time of infection are summarized in Table 1. Overall, the mean age was 55 ± 12 years old, 77 (50%) were female, 55 (36%) had diabetes mellitus (DM), 42 (27%) had hypertension, and the mean BMI was 26.8 ± 3.9 kg/m$^2$. The main causes of cirrhosis that led to LT were HCV infection (n = 44, 29%), metabolic-associated fatty liver disease (n = 26, 17%), and autoimmune hepatitis (n = 19, 12%).

Seventy-nine patients were unvaccinated at the time they developed symptomatic SARS-CoV-2 infection. When comparing vaccinated versus unvaccinated patients, the vaccinated patients were older (mean age of 58 vs. 53 years old, p = 0.007), with female predominance (59% vs. 42%, p = 0.029), and higher prevalence of hypertension (35% vs. 20%, p = 0.039). Time since LT (median of 59 months *versus* 59 months, p = 0.8) and BMI (mean of 27.1 kg/m$^2$ versus 26.5 kg/m$^2$, p = 0.4) were similar between groups.

### 3.2 Immunosuppressive therapy and vaccine types

The immunosuppressive therapy of the LT patients that developed COVID-19 was similar between vaccinated and unvaccinated (Table 2). Most patients had received only one drug (52%), being tacrolimus (95%) the most common, followed by prednisone (35%) and mofetil mycophenolate (25%).

**Table 1. Demographic and clinical characteristics at the time of COVID-19 infection.**

| Characteristic | Unvaccinated | Vaccinated | p-value[2] |
|---|---|---|---|
| | N = 79[1] | N = 74[1] | |
| Age (years) | 53 (11) | 58 (12) | 0.007 |
| Female (%) | 33 (42%) | 44 (59%) | 0.029 |
| BMI (kg/m$^2$) | 27.1 (3.9) | 26.5 (3.9) | 0.4 |
| Etiology (%) | | | 0.3 |
| AIH | 14 (18) | 5 (6.8) | |
| HCV | 20 (25) | 24 (32) | |
| NAFLD | 12 (15) | 14 (19) | |
| Overlap | 4 (5.1) | 3 (4.1) | |
| PSC | 5 (6.3) | 1 (1.4) | |
| PBC | 6 (7.6) | 6 (8.1) | |
| Alcohol | 6 (7.6) | 4 (5.4) | |
| Other | 12 (15) | 17 (23) | |
| Hepatocellular carcinoma | 9 (11%) | 17 (23%) | 0.057 |
| Diabetes | 27 (34%) | 28 (38%) | 0.6 |
| Hypertension | 16 (20%) | 26 (35%) | 0.039 |
| Smoking | 1 (1.3%) | 0 (0%) | >0.9 |
| Time since transplant in months. (range) | 59 (34, 84) | 59 (31, 86) | 0.8 |

[1] Mean (SD); n (%)

[2] t-test; Pearson's Chi-squared test; Fisher's exact test.

**Table 2. Immunosuppressive therapy at the time of COVID-19 infection.**

| Characteristic | Unvaccinated, N = 79[1] | Vaccinated, N = 74[1] | p-value[2] |
|---|---|---|---|
| Prednisone | 32 (41) | 22 (30) | 0.2 |
| Tacrolimus | 73 (92) | 72 (97) | 0.3 |
| Cyclosporine | 3 (3.8) | 1 (1.4) | 0.6 |
| MMF | 17 (22) | 21 (28) | 0.3 |
| Sirolimus | 2 (2.5) | 0 (0) | 0.5 |
| Number of drugs | | | 0.7 |
| 1 | 39 (49) | 41 (55) | |
| 2 | 31 (39) | 24 (32) | |
| 3 | 9 (11) | 9 (12) | |

[1] Mean (SD); n (%)

[2] t-test; Pearson's Chi-squared test; Fisher's exact test.

**Table 3. Vaccine types used in liver transplant recipients.**

| Vaccine received | N = 74[1] |
|---|---|
| BNT162b2 | 36 (48.6) |
| ChAdOx1 nCoV-19 | 16 (21.6) |
| mRNA-1273 vaccine | 1 (1.4) |
| Sputnik V | 11 (14.9) |
| Ad5-nCoV-S | 3 (4.1) |
| CoronaVac | 7 (9.5) |

[1] n (%)

The most frequent vaccines brands among vaccinated LT patients who developed COVID-19 were BNT162b2 (n = 36, 49%), ChAdOx1 nCoV-19 (n = 16, 22%), and Sputnik V (n = 11, 15%). There were 3 patients (4%) with Ad5-nCoV-S, a 1-dose scheme. The complete distribution of vaccines brands received is presented in Table 3.

## 3.3 COVID-19 outcomes: Death due to COVID-19, progression to severe disease and ICU requirement

The relationship between vaccination and COVID-19 outcomes is summarized in Table 4. A total of seven deaths due to COVID-19 occurred among LT patients who developed COVID-19 corresponding to a case fatality of 4.6%. The case fatality was 4.1% (3/74) in vaccinated patients and 5.1% (4/79) in unvaccinated ones, without being these case fatalities significantly different (adjusted relative case fatality for vaccinated versus unvaccinated of 0.68, 95% confidence interval [CI] 0.14–3.24, p = 0.62). Regarding the vaccinated patients who died, three were vaccinated with ChAdOx1 nCoV-19 and one with BNT162b2.

The risk of developing severe COVID-19 was 19% (n = 29) overall, 9.5% (7/74) in vaccinated patients and 28% (22/79) in unvaccinated ones, being this risk significantly lower in the vaccinated group (adjusted relative risk [aRR] of developing severe disease for vaccinated versus unvaccinated of 0.32, 0.14–0.71, p = 0.005). The risk of ICU requirement was 5.9% (n = 9) overall, 4.1% (3/74) in the vaccinated group, and 7.6% (6/79) among unvaccinated patients,

**Table 4. Risk of severe COVID-19, ICU requirement, and in-hospital death among vaccinated and unvaccinated LT patients.**

| Outcome | Unadjusted analysis† | | | | Adjusted analysis¥ | | |
| --- | --- | --- | --- | --- | --- | --- | --- |
| | Risk (%) | RR | 95% CI | P-value | aRR | 95% CI | P-value |
| *Death due to COVID-19* | | | | | | | |
| Vaccinated | 4.1% | 0.80 | 0.19–3.46 | 0.73 | 0.62 | 0.13–3.04 | 0.56 |
| Unvaccinated | 5.1% | | | | | | |
| *Severe COVID* | | | | | | | |
| Vaccinated | 9.5% | 0.34 | 0.15–0.75 | 0.007 | 0.29 | 0.13–0.66 | 0.003 |
| Unvaccinated | 28% | | | | | | |
| *ICU requirement* | | | | | | | |
| Vaccinated | 4.1% | 0.53 | 0.14–2.06 | 0.36 | 0.42 | 0.10–1.77 | 0.24 |
| Unvaccinated | 7.6% | | | | | | |

†Results obtained from a Poisson regression model with vaccination status as the exposure of interest and using robust standard error estimates.

¥ Results obtained from a Poisson regression model with vaccination status as the exposure of interest, adjusting for age and time on liver transplantation shorter than 6 months, and using robust standard error estimates. (ICU: Intensive care unit, LT: liver transplant, RR: relative risk, aRR: adjusted relative risk, CI: confidence interval).

without being these risks significantly different (aRR of UCI requirement for vaccinated versus unvaccinated of 0.45, 95% CI 0.11–1.88, p = 0.27).

The vaccination coverage and COVID-19 cases over time, including their severity, are displayed in Fig 1. Initially, a high incidence of severe illness was noted among LT patients, but as vaccination increased, the number of severe cases decreased and mild cases rose, leading to an inversion in case severity.

## 4. Discussion

In this cohort of LT patients who developed symptomatic SARS-CoV-2 infection, we found that full vaccination is associated with a lower risk of severe disease. However, we did not find statistically significant evidence that COVID-19 vaccination decreases the likelihood of mortality or ICU admission. This observation could be attributed to a lack of statistical power, given the low number of deaths or individuals requiring ICU admission. Our results contrast with those reported by Moon et al. [17] who reported no fatalities among individuals who received a complete vaccine regimen. However, we believe that this discrepancy may be explained by several sample variability (our sample size was 8-fold larger) and due to differences in vaccination protocols and disparities in vaccine efficacy.

The vaccination program in Mexico started in late December 2020 [18]. Healthcare personnel were given priority access to vaccines from the beginning until February 2021. Following this, an age-based approach was employed, with individuals aged 60 and above having access to vaccination between February and May 2020. Those between 50 and 59 years old, as well as pregnant women, had access between May and June 2020. Individuals aged 40 to 49 had access to vaccination between June and July 2020, while the rest of the population received access between July 2020 and March 2022. The specific vaccines administered varied across regions and were contingent upon the availability of a suitable cold chain [18]. In Mexico, immuno-suppressed patients did not receive priority vaccination.

Research in SARS-CoV-2 infection has consistently demonstrated that mRNA vaccines provide a greater humoral response compared to viral vector vaccines in the regular population and LT recipients, where humoral response is measured with antibody titles [14–16, 19]; however, studies focused on major outcomes (developing severe disease, death, etc.) by vaccine

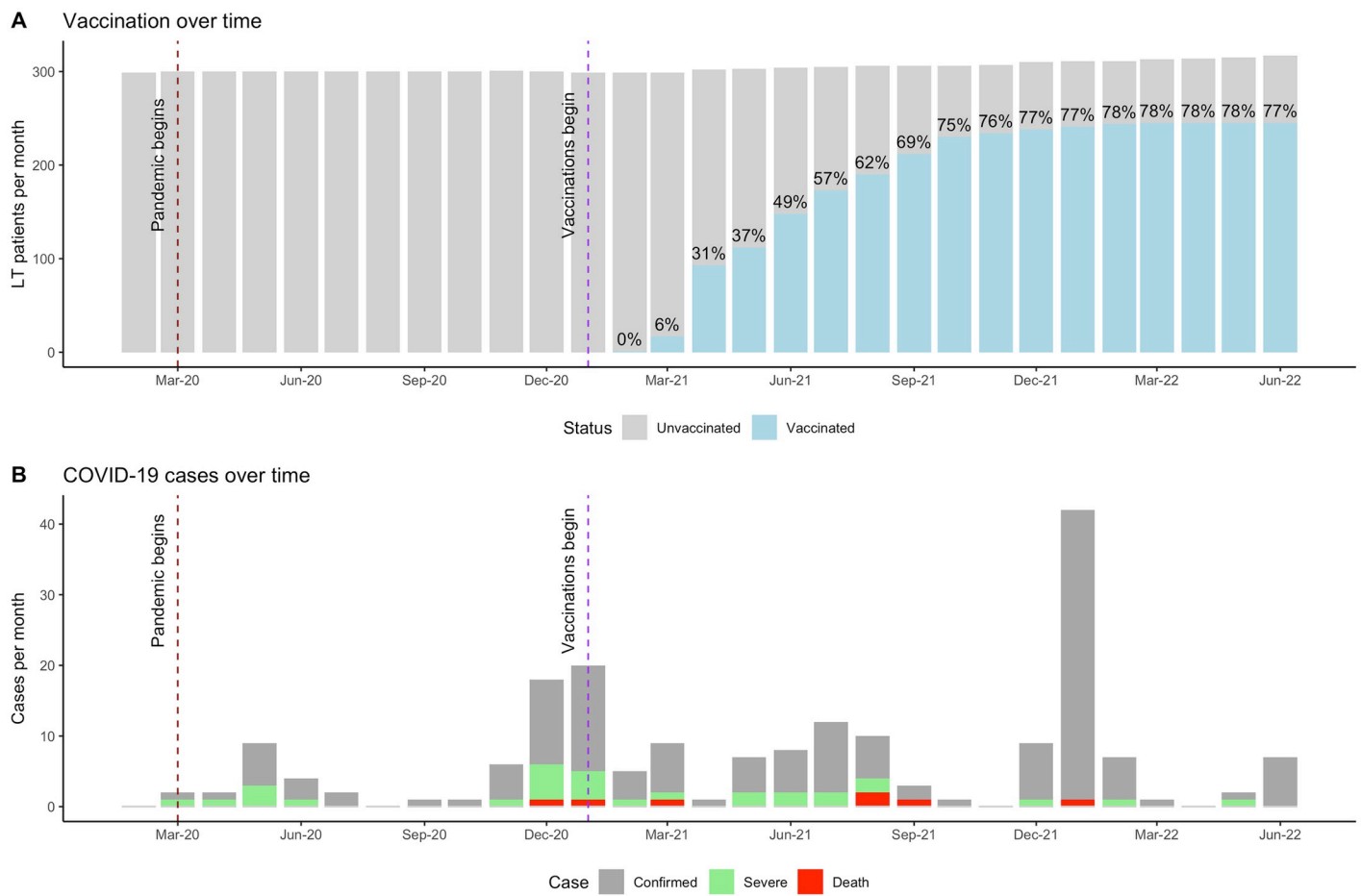

**Fig 1. Vaccination coverage and COVID-19 outcomes over time.** Panel A. Gray bars represent the number of LT patients over time that received attention at our reference center, while blue lines represent the number of vaccinated LT patients over time. Panel B. Gray bars represent the number of COVID-19 confirmed cases, green bars the number of severe cases, and red bars the number of deaths due to COVID-19. This figure shows that as the vaccination coverage increased and that the ratio of severe versus non-severe cases decreased. (LT: liver transplant).

type in LT recipients are scarce. In a study of 1,924 LT recipients that compared BNT162b2 or 1273 mRNA vaccinated and unvaccinated patients, immunization showed a protective effect for symptomatic COVID-19 (aHR 0.42 (0.27–0.65, p>0.0001) and for COVID-19-related death (aHR 0.13 (0.04–0.37) p = 0.0002) [20].

Our study suggests that vaccination through any approved regimen protects against severe illness, even though liver transplant (LT) recipients exhibit a reduced immune response to the vaccine [14–16]. Further investigation is necessary to elucidate the humoral response of non-mRNA-based vaccines.

Overall, the available evidence suggests that COVID-19 outcomes in LT recipients are similar than those in the general population. However, in some reports LT recipients have a higher rate of admission to the intensive care unit. Recent multicenter cohort studies have shown that mRNA vaccines significantly reduce the rates of SARS-CoV-2 infection, symptomatic COVID-19, and death in patients with cirrhosis, including those with LT [21].

A matched cohort of 2307 solid organ transplant recipients (of whom 240 were LT recipients) reported higher COVID-19-related fatality for the solid organ transplant group. However, propensity/matched analyses revealed that this increased risk was secondary to the

higher burden of comorbidities, and after controlling for this burden of comorbidities no difference in intubation or mechanical ventilation was found [21, 22]. Other reports confirmed that age and other comorbidities had a more impact on the outcomes than that conveyed by LT itself [5, 6, 23].

In our study, the case fatality was low (4.6%); and we believe that this may be explained by the low burden of comorbidities in our cohort (<30% had Diabetes or Hypertension), as well as a short time on LT (after the critical period of 6 months).

The literature on the impact of vaccination on LT recipients is scarce. In a study including 29 LT recipients, Hardgrave et al. found that patients who received a complete vaccination scheme had a significantly lower 60-day case fatality following COVID-19 infection compared to unvaccinated patients (11.2% vs. 2.2%) [24]. In the same study, kidney transplant recipients had a significantly higher odds ratio for 60-day case fatality following COVID-19 infection compared to LT recipients (p < 0.001), which may be attributed to differences in immunosuppression regimen or to a higher prevalence of metabolic and cardiovascular comorbidities in kidney transplant recipients. In a retrospective study in Turkey, COVID-19 vaccination was associated with a 100-fold reduction in mortality in LT recipients; in this study, comorbidities such as diabetes mellitus and hypertension were not associated with a higher mortality [25].

We decided not to explore the effect of immunosuppression on patient outcomes and vaccine effectiveness in our analysis because the immunosuppression schemes were very heterogeneous in our cohort and because the sample size is not large enough to assess effect modification. Colmenero et al. [5] described an association between MMF and severe disease, and that the effectiveness of vaccines in producing a humoral response is diminished using MMF in LT patients without prior COVID-19 infection; however, their analysis was exploratory, and their regression models adjusted for more covariates than their sample size can support [26]. Furthermore, these associations had not been consistently reported [6, 17, 25].

On the other hand, a protective role of tacrolimus was advocated in the European experience of Belli et al. [27], where the use of tacrolimus in the immunosuppressive regimen had a positive effect on survival (HR 0.55; CI 95% 0.31 to 0.99). This was also found in a study from Turkey with 387 LT recipients with confirmed COVID, and tacrolimus was found to be a protective factor against death due to COVID 19 (HR 3.44 (95% CI = 1.35–8.33)). They also found that immunosuppression with everolimus (HR 2.94 (95% CI = 1.35–6.42)) and prednisolone (2.53 fold (95% CI = 1.01–6.06)) was associated with a higher mortality; they found no effect on MMF [25]. In our study, 95% of the patients had an immunosuppression regimen with tacrolimus, and 52% were on a single drug scheme with tacrolimus, however we were not able to explore the role of tacrolimus on COVID-19 outcomes. In addition, due to local practice patterns during the period of heightened disease the patients were often treated with steroids alone. This is certainly an area of increasing concern that needs further research to guide immunosuppressive therapy during symptomatic COVID 19 [5, 25, 28].

It is essential to mention that up to 22% of the population was not vaccinated. Even though we considered this number alarmingly high, it is lower than numbers reported in other populations [20]. This could be explained by the age-based prioritization scheme without consideration of the immunocompromised status established by the Mexican government [18]. Another theory could be the adverse effect concerns of newly developed vaccines during the pandemic despite offering good protection against severe disease and death in the general population.

## 4.1 Limitations

This is a single-center study carried out in the hospital that performs most of the LT in Mexico, hence, these findings may not represent the impact of COVID-19 vaccines on LT patients

treated at other centers. This is also a retrospective study, which prevented us from collecting other useful data such that antibody titles, COVID-19 variant and/or subvariant or from administering the same vaccine brand to all the LT patients attended at our center. The most important limitation that applies to any study, even for a randomized clinical trial, is that by focusing only on individuals who got symptomatic SARS-CoV-2 infection, the vaccinated and unvaccinated groups are not comparable. That is, subjects who get the infection in the vaccinated group have on average a worse immune response than those who get the infection on the unvaccinated groups since they were infected even after receiving the immunization. This could also explain the null effect of the COVID-19 vaccines on the post-infection outcomes of ICU requirement and death due to COVID-19. Due to the nature of the study, it is not possible to attribute causality to the findings. We attempt to control for potential confounders, however, due to the small number of mortality events we could not adjust the regression models for more than two variables. And there is also the possibility of remaining confounding since this is not a randomized trial. Furthermore, the effect of the vaccination program and medical tourism was not assessed, which increased the heterogenicity of the study. Finally, we did not include time since vaccination, booster status, and type of vaccine for the analysis.

In conclusion, this analysis suggests that vaccination reduces the risk of severe COVID-19 in LT patients, regardless of the scheme used. Vaccination should be encouraged for all LT recipients.

## Acknowledgments

We would like to extend our sincere gratitude to Silvia López-Yáñez, Victor M Paez-Sayaz, and Jonathan Aguirre-Valadez for their valuable contribution to the data collection process.

## Author Contributions

**Conceptualization:** Daniel Azamar-Llamas, Antonio Olivas-Martinez, Eric Kauffman-Ortega, Ignacio García-Juárez.

**Data curation:** Daniel Azamar-Llamas, Josealberto Sebastiano Arenas-Martinez, Antonio Olivas-Martinez, Cristian J García-Carrera, Bruno Papacristofilou-Riebeling, Fabián E Rivera-López, Ignacio García-Juárez.

**Formal analysis:** Daniel Azamar-Llamas, Josealberto Sebastiano Arenas-Martinez, Ignacio García-Juárez.

**Investigation:** Josealberto Sebastiano Arenas-Martinez, Eric Kauffman-Ortega, Cristian J García-Carrera, Bruno Papacristofilou-Riebeling, Fabián E Rivera-López, Ignacio García-Juárez.

**Methodology:** Daniel Azamar-Llamas, Josealberto Sebastiano Arenas-Martinez, Antonio Olivas-Martinez, Jose Victor Jimenez, Ignacio García-Juárez.

**Supervision:** Antonio Olivas-Martinez.

**Validation:** Josealberto Sebastiano Arenas-Martinez, Antonio Olivas-Martinez.

**Writing – original draft:** Daniel Azamar-Llamas, Josealberto Sebastiano Arenas-Martinez, Antonio Olivas-Martinez, Ignacio García-Juárez.

**Writing – review & editing:** Daniel Azamar-Llamas, Josealberto Sebastiano Arenas-Martinez, Antonio Olivas-Martinez, Jose Victor Jimenez, Ignacio García-Juárez.

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
