## [Decision Letter · Decision Letter 0]

4 Jan 2024

PONE-D-23-25428Impact of COVID-19 vaccination on liver transplant recipients. Experience in a reference center in Mexico.PLOS ONE

Dear Dr. Ignacio García-Juárez,

Thank you for submitting your manuscript to PLOS ONE. After careful consideration, we feel that it has merit but does not fully meet PLOS ONE’s publication criteria as it currently stands. Therefore, we invite you to submit a revised version of the manuscript that addresses the points raised during the review process.

We look forward to receiving your revised manuscript.

Kind regards,

Fadi Aljamaan

Academic Editor

PLOS ONE

Journal Requirements:

3. In this instance it seems there may be acceptable restrictions in place that prevent the public sharing of your minimal data. However, in line with our goal of ensuring long-term data availability to all interested researchers, PLOS’ Data Policy states that authors cannot be the sole named individuals responsible for ensuring data access (http://journals.plos.org/plosone/s/data-availability#loc-acceptable-data-sharing-methods).

Reviewers' comments:

Reviewer's Responses to Questions

**Comments to the Author**

1. Is the manuscript technically sound, and do the data support the conclusions?

Reviewer #1: Partly

Reviewer #2: Partly

2. Has the statistical analysis been performed appropriately and rigorously? 

Reviewer #1: Yes

Reviewer #2: Yes

3. Have the authors made all data underlying the findings in their manuscript fully available?

Reviewer #1: Yes

Reviewer #2: Yes

4. Is the manuscript presented in an intelligible fashion and written in standard English?

Reviewer #1: No

Reviewer #2: Yes

5. Review Comments to the Author

Reviewer #1: I was very happy because the topic presented in the article was from an area that I was specifically interested in.

Authors ''The risk in liver transplant (LT) patients

"has not been widely investigated," they said. This statement is unrealistic in my opinion. There are very serious studies written on this subject.

Please use the following studies to enrich the introduction and discussion sections of your article:

Akbulut S, Yagin FH, Sahin TT, Garzali IU, Tuncer A, Akyuz M, Bagci N, Barut B, Unsal S, Sarici KB, Saritas S, Ozer A, Bentli R, Colak C, Bayindir Y, Yilmaz S. Effect of COVID-19 Pandemic on Patients Who Have Undergone Liver Transplantation: Retrospective Cohort Study. J Clin Med. 2023;12(13):4466.

Akbulut S, Bagci N, Akyuz M, Garzali IU, Saritas H, Tamer M, Ince V, Unsal S, Aloun A, Yilmaz S. Effect of COVID-19 Pandemic on Patients Who Have Undergone Liver Transplantation Because of Hepatocellular Carcinoma. TransplantProc. 2023;55(5):1226-1230.

Akbulut S, Barut B, Garzali IU, Sarici KB, Tamer M, Unsal S, Karabulut E, Baskiran A, Bayindir Y, Yilmaz S. Effect of Pre-Transplant Covid-19 Exposure on Post-Liver Transplant Clinical Outcomes. TransplantProc. 2023;55(5):1176-1181.

Akbulut S, Sahin TT, Ince V, Yilmaz S. Impact of COVID-19 pandemic on clinicopathological features of transplant recipients with hepatocellular carcinoma: A case-control study. World J Clin Cases. 2022;10(15):4785-4798.

Sahin TT, Akbulut S, Yilmaz S. COVID-19 pandemic: Its impact on liver disease and liver transplantation. World J Gastroenterol. 2020; 26(22):2987-2999.

Delete the "Overall" column in the tables. Leave only the columns belonging to the two groups and the column where the "p" value is given in the table.

In the latest study published by Akbulut et al. (PMID: 37445501; J Clin Med. 2023 Jul 3;12(13):4466.), the factors affecting mortality in atransplant patients were seriously examined and it was stated that vaccination reduced mortality by 100 times. In the same study, immunosuppressive drugs and comorbidities were shown to be associated with mortality. It is very important to comment on this study and other large studies like it in the discussion section.

Please add a paragraph about the limitations of the study to the last part of the discussion.

Reviewer #2: The authors assessed the impact of COVID-19 vaccination on the mortality and development of severe and critical disease in LT recipients. They found that vaccination reduces the risk of severe COVID-19 in LT patients, regardless of the scheme used. I think this conclusion is well acknowledged. So this topic seems lack of novelty to the readers of Plos One Journal.

6. PLOS authors have the option to publish the peer review history of their article (what does this mean?). If published, this will include your full peer review and any attached files.

Reviewer #1: No

Reviewer #2: No

---

## [Author Response · Author response to Decision Letter 0]

14 Jan 2024

Reviewer #1: 

1. I was very happy because the topic presented in the article was from an area that I was specifically interested in.

Authors ''The risk in liver transplant (LT) patients

"has not been widely investigated," they said. This statement is unrealistic in my opinion. There are very serious studies written on this subject.

a. The paragraph was changed to acknowledge research that was recently published.

i. Background and aims: COVID-19 vaccination has proven to be effective in preventing symptomatic infection and severe disease, even in immunocompromised patients, including liver transplant patients. We aimed to assess the impact of the COVID-19 vaccination on mortality and the development of severe and critical disease in our center. 

2. Please use the following studies to enrich the introduction and discussion sections of your article:

Akbulut S, Yagin FH, Sahin TT, Garzali IU, Tuncer A, Akyuz M, Bagci N, Barut B, Unsal S, Sarici KB, Saritas S, Ozer A, Bentli R, Colak C, Bayindir Y, Yilmaz S. Effect of COVID-19 Pandemic on Patients Who Have Undergone Liver Transplantation: Retrospective Cohort Study. J Clin Med. 2023;12(13):4466.

a. This article has been cited in the Discussion section (reference number 25).

i. This was also found in a study from Turkey with 387 LT recipients with confirmed COVID, and tacrolimus was found to be a protective factor against death due to COVID 19 (HR 3.44 (95% CI = 1.35–8.33)). They also found that immunosuppression with everolimus (HR 2.94 (95% CI = 1.35–6.42)) and prednisolone (2.53 fold (95% CI = 1.01–6.06)) was associated with a higher mortality; they found no effect on MMF [24]. 

Akbulut S, Bagci N, Akyuz M, Garzali IU, Saritas H, Tamer M, Ince V, Unsal S, Aloun A, Yilmaz S. Effect of COVID-19 Pandemic on Patients Who Have Undergone Liver Transplantation Because of Hepatocellular Carcinoma. TransplantProc. 2023;55(5):1226-1230.

a. This article has been cited in the Introduction section (reference number 8).

i. Liver transplant (LT) patients do not appear to be more prone to COVID-19 infection, with a similar incidence rate as the general population (3.18 cases/100 person-years in LT patients vs. 3.97 cases/100 person-years in non-LT patients), also featuring lower in-hospital mortality (18% vs. 27%) [5–8].

In response to the reviewer's feedback, we thoroughly examined the other references suggested for our paper. However, upon careful analysis, we concluded that these references were not pertinent or directly relevant to the content and focus of our paper. It's crucial to note that the decision was based on a comprehensive evaluation of the suggested sources in relation to the specific objectives, scope, and context of our research.

3. Delete the "Overall" column in the tables. Leave only the columns belonging to the two groups and the column where the "p" value is given in the table.

a. The overall column has been deleted from all the tables.

4. In the latest study published by Akbulut et al. (PMID: 37445501; J Clin Med. 2023 Jul 3;12(13):4466.), the factors affecting mortality in atransplant patients were seriously examined and it was stated that vaccination reduced mortality by 100 times. In the same study, immunosuppressive drugs and comorbidities were shown to be associated with mortality. It is very important to comment on this study and other large studies like it in the discussion section.

a. The following paragraph has been added to present the information demonstrated in this study.

i. In a retrospective study in Turkey, COVID-19 vaccination was associated with a 100-fold reduction in mortality in LT recipients; in this study, comorbidities such as diabetes mellitus and hypertension were not associated with higher mortality [25]. 

ii. This was also found in a study from Turkey with 387 LT recipients with confirmed COVID, and tacrolimus was found to be a protective factor against death due to COVID 19 (HR 3.44 (95% CI = 1.35–8.33)). They also found that immunosuppression with everolimus (HR 2.94 (95% CI = 1.35–6.42)) and prednisolone (2.53 fold (95% CI = 1.01–6.06)) was associated with a higher mortality; they found no effect on MMF [25].

5. Please add a paragraph about the limitations of the study to the last part of the discussion.

a. We acknowledge our limitations and have added a subtitle to this part of the discussion.

i. We performed a single center study with the limitations that it carries like a limited number of patients. We do not know the COVID variant that these patients were infected with or their antibody titers. We attempted to control for potential confounders; however, because of the low mortality rate, we adjusted the regression models for only two variables. 

Reviewer #2: 

1. The authors assessed the impact of COVID-19 vaccination on the mortality and development of severe and critical disease in LT recipients. They found that vaccination reduces the risk of severe COVID-19 in LT patients, regardless of the scheme used. I think this conclusion is well acknowledged. So this topic seems lack of novelty to the readers of Plos One Journal.

a. We appreciate the input by the reviewer; however, we think that this study is of relevance because the study shows quality data of a high-volume reference center of liver transplant in a previously.

---

## [Decision Letter · Decision Letter 1]

13 Mar 2024

Impact of COVID-19 vaccination on liver transplant recipients. Experience in a reference center in Mexico.

PONE-D-23-25428R1

Dear Dr. Ignacio García-Juárez,

We’re pleased to inform you that your manuscript has been judged scientifically suitable for publication and will be formally accepted for publication once it meets all outstanding technical requirements.

Kind regards,

Fadi Aljamaan

Academic Editor

PLOS ONE

Additional Editor Comments (optional):

Reviewers' comments:

Reviewer's Responses to Questions

**Comments to the Author**

1. If the authors have adequately addressed your comments raised in a previous round of review and you feel that this manuscript is now acceptable for publication, you may indicate that here to bypass the “Comments to the Author” section, enter your conflict of interest statement in the “Confidential to Editor” section, and submit your "Accept" recommendation.

Reviewer #1: All comments have been addressed

Reviewer #2: All comments have been addressed

2. Is the manuscript technically sound, and do the data support the conclusions?

Reviewer #1: Yes

Reviewer #2: Yes

3. Has the statistical analysis been performed appropriately and rigorously? 

Reviewer #1: Yes

Reviewer #2: Yes

4. Have the authors made all data underlying the findings in their manuscript fully available?

Reviewer #1: Yes

Reviewer #2: Yes

5. Is the manuscript presented in an intelligible fashion and written in standard English?

Reviewer #1: (No Response)

Reviewer #2: Yes

6. Review Comments to the Author

Reviewer #1: I saw the changes made to the article. I think the article can be published in its current form.

Best regards

Reviewer #2: The authors have addressed my concerns. I don't have further questions. And I think this article could be accepted to publish in Plos one

7. PLOS authors have the option to publish the peer review history of their article (what does this mean?). If published, this will include your full peer review and any attached files.

Reviewer #1: No

Reviewer #2: No

---

## [Editor Report · Acceptance letter]

19 Mar 2024

PONE-D-23-25428R1 

PLOS ONE

Dear Dr. García-Juárez, 

I'm pleased to inform you that your manuscript has been deemed suitable for publication in PLOS ONE. Congratulations! Your manuscript is now being handed over to our production team.

Kind regards, 

on behalf of

Dr. Fadi Aljamaan 

Academic Editor

PLOS ONE